# Novel Thermoplastic Composites Strengthened with Carbon Fiber-Reinforced Epoxy Composite Waste Rods: Development and Characterization

**DOI:** 10.3390/polym14193951

**Published:** 2022-09-21

**Authors:** José Antonio Butenegro, Mohsen Bahrami, Yentl Swolfs, Jan Ivens, Miguel Ángel Martínez, Juana Abenojar

**Affiliations:** 1Materials Science and Engineering and Chemical Engineering Department, IAAB, University Carlos III Madrid, 28911 Leganés, Spain; 2Department of Materials Engineering, KU Leuven, Kasteelpark Arenberg 44, B-3001 Leuven, Belgium; 3Mechanical Engineering Department, Universidad Pontificia Comillas, Alberto Aguilera 25, 28015 Madrid, Spain

**Keywords:** polymer composites, carbon fiber-reinforced polymers, thermoplastic polymers, recycling, properties

## Abstract

The increasing use of carbon fiber and epoxy resin composite materials yields an increase in the amount of waste. Therefore, we present a solution consisting of composites manufactured by hot pressing, employing polyamides (either PA11 or PA12) and a mechanically recycled carbon fiber-reinforced polymer (CFRP) as reinforcement. The main objectives are to study the manufacturing of those composites, to evaluate the fiber distribution, and to perform a mechanical, dynamical, and thermomechanical characterizations. The X-ray micro-computed tomography (μCT) shows that the fibers are well-distributed, maintaining a homogeneous fiber volume fraction across the material. The variability in the results is typical of discontinuous fiber composites in which the fibers, although oriented, are not as homogeneously distributed as in a continuous fiber composite. The mechanical and dynamic properties barely differ between the two sets of composites. A dynamic-mechanical analysis revealed that the glass transition temperature (T_g_) increases slightly for both composites, compared to the polymers. These results illustrate the viability of the recycling and reuse route for preventing the deterioration of carbon fibers and promoting the subsequent reduction in the environmental impact by employing a thermoplastic matrix.

## 1. Introduction

The demand for carbon fiber-reinforced polymer (CFRP) composite materials due to their lightweight and high performance has swiftly increased over the past years, which has been mainly driven by the aerospace, automotive, defense, wind turbine, marine, leisure, and construction industries [1,2,3,4,5,6]. The rapid increase in the demand for CFRP composite materials is motivated primarily by their high specific strength and specific stiffness, high fatigue resistance and durability, good corrosion resistance, and low density. However, the recycling of fiber-reinforced polymer (FRP) composite materials is one of their most critical disadvantages. FRP recycling is not motivated strictly by economic reasons; moreover, the objectives of obtaining recycled fibers and reusing or repurposing FRPs at their end-of-life stage lie in minimizing the volume of FRP waste that is expected to increase in the coming years [7].

Nowadays, the central challenges in composite materials development are industry competitiveness, capacity, and sustainability. The need for recycling these materials is aligned with the objectives of the circular economy, in order to make manufacturing and recycling processes more efficient while minimizing the waste of CFRP composites at their end-of-life [7,8]. The strategies for minimizing the environmental impact of composite materials address a number of key issues, including the development of new materials using recycled carbon fibers, natural fibers, or bio-based polymers [9,10,11]. On the matrix side, thermoplastic matrices are also employed. In contrast, thermosetting matrices have been generally used until now, particularly epoxy, polyester, and acrylic resins. A trade-off is present in terms of properties since thermosetting polymer matrices typically have better mechanical properties and exhibit better fiber–matrix adhesion compared to thermoplastic matrices. Bahrami et al. [12] investigated the influence of plasma treatment on the fiber–matrix adhesion in carbon fiber and hybrid carbon/flax fiber-reinforced composites, employing PA11 and PA12 as matrices. They found an increase in wettability, polar surface energy, and adhesion bonding due to the plasma treatment of PA11 and PA12 surfaces.

Concerning the selection of thermoplastic matrices, two Z-type polyamides, PA11 and PA12, were selected in this study. Even though PA6 and PA66 account for almost 90% of the polyamide market, PA11 and PA12 also have a significant market share. PA11 is a bio-based polyamide obtained from castor oil, a renewable source, thus resulting in a reduced carbon footprint compared to PA12 [13,14]. Apart from being bio-based, PA11 is experiencing a sharp increase in demand due to several of its features, including its low moisture absorption and high impact, fatigue, chemical, and aging resistance [15]. PA12 is an oil-based thermoplastic polyamide with outstanding mechanical properties, including tensile strength, impact and fatigue resistance, resistance to aromatic hydrocarbons, and a low coefficient of friction close to those of PA6 and PA66 [16]. In addition, both PA11 and PA12 have a relatively low melting temperature (around 40–50 °C below PA6 and 90–100 °C below PA66), reducing energy consumption and enabling the manufacture of composites using natural fibers [17,18].

A cost-effective approach to reducing environmental impacts is the mechanical recycling of carbon fibers, which avoids the issues related to scalability or to the use and disposal of the chemicals required for chemical recycling and reduces the high energy inputs required for thermal recycling. Mechanical recycling also avoids problems related to the emission of pollutant gases or having to deal with solvents’ disposal. Hence, mechanical recycling makes it possible to obtain carbon fibers or composites in the shape of small rods of different lengths. To preserve the mechanical properties of carbon fibers, the objective is to obtain a reinforcement consisting of long fibers, defined as those that retain properties similar to the properties of continuous fibers [19]. Even though obtaining short carbon fibers by mechanical means and manufacturing composites with them is relatively easier compared to using longer carbon fibers, researchers’ interest has been limited. As for using a mechanically recycled CFRP as reinforcement, there is hardly any literature on the subject.

The mechanical and physical properties of composites highly depend on the fiber volume fraction and the fiber orientation distribution. X-ray micro-computed tomography (μCT) is useful for analyzing the fiber volume distribution inside the composites. Several authors have used μCT in composites to determine their fiber orientation distribution [20,21,22] and their fiber misalignment [23]; to analyze void characteristics [24]; to determine the effect of fiber length, diameter, and orientation [25]; or to achieve a non-contact characterization of deformation and damage [26]. Yu et al. [27] studied the variation in the yarn fiber volume fraction induced by the compression between adjacent yarns during the manufacturing process of textile composites. They leveraged a combination of SEM and μCT to enhance their analysis as follows: they measured the fiber cross-sectional area by means of SEM and used μCT to measure the variation in the yarn’s cross-sectional area at the mesoscale. This approach allowed them to characterize their materials, decomposing the variations into a systematic trend and a stochastic deviation. Wan and Takahashi [28] manufactured ultra-thin chopped carbon fiber tape composites using polyamide 6 as a matrix by compression molding. The composite consisted of a randomly oriented discontinuous fiber system, with a set of reinforcement lengths of 6 to 24 mm and a width of 5 to 6 mm. They found that the tensile and compressive modulus values were almost independent of the reinforcement tape lengths and impregnation qualities, but an increase in the tape length led to a slight increase in the structural integrality and therefore the composite’s tensile and compressive strength.

The aim of this publication is the development, manufacture, and characterization of a novel carbon fiber epoxy-reinforced thermoplastic composite. The main contribution of this research is the incorporation of a mechanically recycled CFRP in the shape of rods as fillers or reinforcement within a thermoplastic matrix, without removing the previous thermoset matrix. The characterization comprises the study of the mechanical, dynamic, and thermomechanical behavior of the composites. The characterization is enhanced with μCT to determine the fiber volume distribution inside the composites. This study’s major difference from previous research consists in using thin rods of recycled CFRP, enabling a better wettability during material processing.

## 2. Materials and Methods

### 2.1. Materials

Two commercially available thermoplastic polymeric resins were selected and used as matrices in the shape of pellets, namely, PA11 and PA12, both of which were provided by Arkema (Barcelona, Spain). These polyamides were selected due to their relatively low melting points, which leads to lower energy consumption during production and to their outstanding mechanical properties, including high tensile strength, high impact and fatigue resistance, high chemical and aging resistance, and low moisture absorption [29].

Commercially available pultruded carbon fiber plates were used, which are typically used for structural strengthening in civil engineering. This unidirectional carbon fiber-reinforced epoxy composite, with the trade name Sika Carbodur S 512 and from now on referred to as Carbodur S512, was provided by Sika S.A.U. España (Alcobendas-Madrid, Spain). The Carbodur S512 plates of 1.2 mm thickness were mechanically cut and reduced to rods of 40 mm length and variable widths, specifically, between 1 mm and 1.5 mm. Therefore, these rods are representative of the type of material that would result from mechanical recycling.

Table 1 presents several properties, including thermal and mechanical properties of both thermoplastic polymers (PA11 and PA12) and the reinforcement (Carbodur S512).

Table 2 summarizes the fiber volume fraction and thickness, as well as representative mechanical properties of Carbodur S512.

### 2.2. Manufacturing of Specimens

The manufacturing process was carried out by means of a hydraulic press (Fontijne Presses TPB374, Barendrecht, The Netherlands). First, the polyamide pellets were hot-pressed without previous drying between aluminum plates, which was conducted inside a steel frame to obtain a thin sheet that would later serve as a matrix. Then, the Carbodur S512 was cut in the shape of rods 40 mm long to serve as fillers or reinforcement. A total of 50 g of reinforcement was manually placed in an equally distributed manner on the top and on the bottom of the manufactured polyamide sheet, between aluminum plates and inside a steel frame, and hot-pressed again to manufacture the final composite sheet. The dimensions of the final composite sheets were the following (in mm): length (L_CFRP_) = 280 ± 1, width (W_CFRP_) = 280 ± 1, and thickness (T_CFRP_) = 1.3 ± 0.1. The edges of the composite sheets were discarded, as these contained only a matrix. The final weight of the composite plates was 105 ± 2 g. Figure 1 exhibits CarbodurS512 before and after cutting and the final manufactured composites.

Figure 2 depicts the heating and pressure cycles applied for manufacturing both the polyamide sheets. The cycle was very similar for the final composite sheets.

The reinforcement thickness practically coincides with the thickness of the fabricated plates, so the present case may correspond to that of a flat structure. The estimation of the optimum reinforcement length has been developed experimentally, yielding 40 mm as optimum, which may lead to differences with shear-lag model, where unidirectional reinforcements are considered [34]. This ideal one-dimensional nature cannot be fully achieved in this case due to the deflection of part of the reinforcements during pressing, a situation beyond our control.

The different specimens for the various tests were then obtained by waterjet cutting. Table 3 summarizes the dimensions for the different tests.

After preparation, all specimens were stored under controlled humidity and temperature conditions (23 °C and 50% RH) until testing, as both polyamides and their composites absorb water within a few minutes of exposure to humid environments [35,36].

### 2.3. Determination of Fiber Volume Fraction

#### 2.3.1. A Theoretical Approach: The Rule of Mixtures in Composites

The fiber volume fraction (V_f_) of a composite is a critical parameter in composites’ behavior and, therefore, in composites’ design. A simple yet valid approximation to the theoretical value of fiber volume fraction is the rule of mixtures, as depicted in Equation (1):
(1)$$V_{f}=\frac{w_{f}\times \rho_{m}}{w_{f}\times \rho_{m}+w_{m}\times \rho_{f}}$$
where w_f_ is the weight of the fibers, w_m_ is the weight of the matrix, ρ_f_ is the density of the fibers, and ρ_m_ is the density of the matrix.

In a similar fashion and more adequate to the present research case, a reinforcement volume fraction (V_r_) should be used instead of V_f_. This leads to Equation (2):(2)$$V_{r}=\frac{w_{r}\times \rho_{m}}{w_{r}\times \rho_{m}+w_{m}\times \rho_{r}}$$
where—instead of referring to the weight, density, and fiber volume of the fibers—the weight and density of the reinforcement as a whole are considered as w_r_ and ρ_r_, respectively, to yield V_r_. Table 1 presents the densities of both matrices and the reinforcement. Although as mentioned before, calculating V_r_ is more convenient for a rod-like reinforced composite. A simple way to calculate the composite V_f_ is to multiply its V_r_ by the reinforcement V_f_, reported in Table 2.

#### 2.3.2. A Practical Approach: X-ray Micro-Computed Tomography

X-ray micro-computed tomography (μCT) was used to analyze the fiber distribution inside the composites [37,38]. An X-ray micro-computed tomography system (TeScan Unitom HR, Ghent, Belgium) was used to study the composites in sets of three specimens, later used in fatigue tests. In total, a set of PA11 composites and another set of PA12 composites were analyzed for a total of six randomly selected specimens. During the image acquisition, each set of three specimens was centered on a holder to be scanned simultaneously. The scans were performed with a tube voltage of 50 kV, a current of 1980 µA, an exposure time of 67 ms, a voxel size of 105 μm, and in the microfocus mode. After image acquisition, the micro-CT images were reconstructed using the Panthera software. To process the images, the ImageJ software was used to convert the original CT images into a three-dimensional array of 8-bit grey values.

The image stack was analyzed by applying Otsu as a threshold in two stages: in the first stage, the reinforcement-to-image ratio is determined, while in the second stage, the composite-to-image ratio is obtained. In this way, an array representing the local V_r_ along the specimen can be obtained as a ratio between the values extracted in the first stage and those obtained in the second stage. This process was performed individually and independently for each specimen. The NumPy and Pandas libraries have been used for the analysis, and Matplotlib and Seaborn for the visualization of the data.

### 2.4. Tensile Testing

Specimens analyzed with digital image correlation techniques needed special preparation before applying the speckle pattern. First, the specimens were ground with sandpaper and cleaned with alcohol. After drying, three layers of white paint were applied. Then, an acrylic resin dispersion in the form of a gel was applied and the speckle patterns were placed on top of the resin, applying a mild pressure to improve contact before letting the gel cure overnight.

Tensile tests were performed in a universal testing machine (Instron 5567, Norwood, MA, USA) following the ISO 527-5 standard, selecting a rate of 1 mm/min, with a distance between grips of 150 mm. Five specimens were tested per set. Sandpaper was used in between the specimens and the grips to avoid slippage. The longitudinal strain was measured by a 2D digital image correlation (DIC) system. Image-processing techniques such as DIC are commonly used in situ for real-time damage analysis. DIC enables the quantification of the deformation produced in the specimens, with a specific focus on localized deformations, thereby allowing for the detection of small-scale damage mechanisms, such as fiber–matrix debonding, matrix cracking and, eventually, fiber breakage [39]. The processing and analysis were carried out using specific software, selecting normalized square differences as the criterion and subsetting weights by Gaussian weights.

Regarding post-processing, a decay filter was used, and a Lagrange tensor was selected. To match the stress values measured by the universal testing machine with the strain values obtained by DIC, the Lagrange strain values obtained in the different subsets had to be averaged and converted. The engineering strain, also known as the Cauchy strain and denoted as ε_ENG_, is defined as follows:(3)$$\varepsilon_{\mathrm{ENG}}=\lambda -1=\frac{L-L_{0}}{L_{0}}$$
where the factor λ corresponds to the ratio between the length and the initial length. The Lagrange strain is defined in relation to λ as:(4)$$\varepsilon_{\mathrm{LAG}}=\frac{1}{2}\left(\lambda^{2}-1\right)$$

By operating with Equations (3) and (4), the engineering strain (ε_ENG_) can be related to the Lagrange strain (ε_LAG_), as seen in Equation (5):(5)$$\varepsilon_{\mathrm{ENG}}=\sqrt{1+2\varepsilon_{\mathrm{LAG}}}-1$$

### 2.5. Fatigue Testing

The tension–tension fatigue tests were performed using a fatigue-testing machine (Instron 8502, Norwood, MA, USA) under load control by applying a sinusoidal load with a frequency of 5 Hz. The stress ratio R, defined as minimum to maximum applied stress, was kept constant and equal to 0.1. In the absence of fracture, tests were stopped after $5\times 10^{5}$ cycles. Fifteen specimens were tested for each matrix at different load levels, covering a broad range of load levels for the composites manufactured with both polyamides.

After the fatigue tests, the specimens were subjected to microscopic observations using a Tagarno FHD Prestige digital microscope (Tagarno, Horsens, Denmark).

### 2.6. Thermomechanical Behavior of Polymers and Composites

The glass transition temperatures (T_g_) of both polymers and composites were measured by a dynamic mechanical analysis (DMA) system (TA Instruments Q800, New Castle, DA, USA). DMA experiments were conducted with a fixed amplitude of 40 μm at a frequency of 1 Hz using a single-cantilever configuration, selecting the multi-frequency strain mode. Table 3 shows the distances between gauges and the specimens’ dimensions. Three specimens were tested for each series: PA11 and PA12 polymers, and PA11 and PA12 composites. Figure 3a exhibits an image of DMA single cantilever clamp, where the medium clamp is adjustable, and the external clamp is fixed. Figure 3b shows a schematic of DMA single-cantilever operation. A heating rate of 3 °C/min was selected to run DMA scans between −40 °C and the melting point of the matrix, with a 5 min isothermal period at −40 °C, according to the ISO 6721-11 standard. The T_g_ was defined, according to the aforementioned standard, as the peak in the loss modulus curve [40]. The loss modulus peak obtained at 1 Hz correlates well with the T_g_ obtained by Differential Scanning Calorimetry (DSC) at 20 K/min. Compared to using the peaks of the storage modulus or the loss factor (tan δ), the loss modulus peak is an intermediate point. The temperature indicated by the inflection point in the storage modulus curve, represented in linear scale, marks the starting point of the glass transition, while the temperature marked by the loss factor peak is influenced by the decrease in the storage modulus and, therefore, is usually higher than that of the loss modulus peak.

## 3. Results and Discussion

### 3.1. Determination of Fiber Volume Fraction

#### 3.1.1. A Theoretical Approach: The Rule of Mixtures in Composites

The V_r_ was obtained for both PA11 and PA12 composites by applying the rule of mixtures in composites formulated in Equation (2). The density values of the matrices and reinforcement and the mass values for the reinforcement and final composites are indicated in Table 1 and Section 2.2, respectively. Table 2 shows that the fiber volume content of Carbodur S512 is higher than 68%; hence, this value was chosen, assuming that it represents the most critical case. By multiplying the V_r_ obtained by applying the rule of mixtures by the fiber volume content reported for Carbodur S512, the V_f_ in the material can be estimated. Table 4 summarizes the theoretical results.

#### 3.1.2. A Practical Approach: X-ray Micro-Computed Tomography

By analyzing the stack of images obtained by the μCT of the cross section of the composites, the V_r_ of the PA11 and PA12 composites were obtained. In the same way, as in the last point of the theoretical approach, the V_f_ can be calculated by simply multiplying the V_r_ by the fiber volume content of Carbodur S512, as reported in Table 2. Table 4 presents the mean and standard deviation for the PA11 and PA12 composites.

Furthermore, a statistical analysis was carried out to study the variability in the distribution at a local level. To this end, four Python libraries were used. First, the data obtained after processing the stack of images with ImageJ was manipulated with Pandas. After cleaning the data, meaningful statistical parameters were calculated with NumPy. Finally, the plots were created and customized with Matplotlib and Seaborn. For the sake of clarity, Figure 4a,b show box plots or box-and-whisker plots. The V_r_ is shown on the y-axis for the six specimens analyzed. The position in the CT-Scan identifies the specimens depending on the position: up, mid, and low, as can be seen in Figure 4c.

As can be seen in Figure 4a,b, outliers are only observed in the upper values. In the whole dataset, no outliers have been found below the lower whisker in any case. Very low values would indicate a low local V_r_, which would indicate that the composite contains an area more prone to a catastrophic failure. Comparing these plots against the data presented in Table 4, it seems that the rule of mixtures slightly overestimates the V_r_ in the case of the PA12 composites. The low standard deviation indicates that the reinforcement rods are quite equally distributed but yields no information regarding the fiber orientation distribution.

### 3.2. Tensile Strength

The mechanical properties of the composites were obtained from the tensile tests combined with DIC. The results for the composites are compared with those obtained for the polymers from the dog-bone specimens in Table 5. To better reflect the deviations present due to the inhomogeneity of the composites, the coefficient of variation (CoV) is reported together with the tensile test results.

Bahrami et al. [29] reported large differences in the tensile behaviors of PA11 and PA12 polymers. While PA11 exhibited a quasi-brittle behavior, PA12 demonstrated a highly ductile behavior, showing a higher tensile strength, Young’s modulus, and strain at failure. They found that the disparity lay in the degree of crystallinity of the polymers, with the tensile strength and Young’s modulus being directly related to the crystal phase of the polymers. In the case of the composites manufactured following the procedure depicted in Figure 2, the differences found in the ultimate tensile strength (UTS) and in the Young’s modulus between the PA11 and PA12 composites are barely noticeable and fall within one standard deviation. However, the behavior is the opposite to that of the polymers in terms of the strain at failure. Although the PA12 polymers show a much more ductile behavior, what is observed in the composites, where the PA11 composites deform substantially more, seems to be related to a relatively superior fiber–matrix adhesion.

Figure 5 presents photographs of the tested specimens. It is particularly clear in the PA12 composites how the fracture advances through the matrix being guided by the reinforcement.

The path followed by the fracture after the tensile test can be noticed in specimens 1, 2, and 4 of the PA12 composites in Figure 5b. It is particularly important to ensure that the reinforcement maintains its one-dimensional nature throughout the material and that the composite rods are uniformly distributed without forming bundles. As a result, the path that the fracture must follow is lengthened, thus improving the mechanical behavior of the composite. Figure 5c shows the failure area of a PA12 composite specimen, illustrating a matrix fracture close to the reinforcement ends. A catastrophic failure happens when these pores and cavities are penetrated by the simplest failure paths, either breaking the matrix or the composite rods longitudinally, causing matrix cracking or intralaminar matrix cracking, respectively.

### 3.3. Fatigue Tests

Figure 6 shows the results obtained after the tension–tension fatigue tests. It is notable that the fatigue strength—even at low cycles—against failure is significantly below the UTS. One hypothesis suggests a strain rate dependence since fatigue tests are performed at much higher rates than tensile tests (approximately 180 mm/min vs. 1 mm/min). A second hypothesis is related to defect sensitivity. The early failure in some of the specimens seems to indicate that the stress concentrations in the material can quickly generate critical cracks. Presumably, the crack propagation is very fast, and the time to failure depends on the initiation time, which is very much related to the local stress concentrations. Further research is needed to explain this drop in strength.

The nearly flat distribution of the values in Figure 6 indicates an almost absolute independence between the stress and the number of cycles prior to failure. A similar behavior was observed in sheet-molding compounds, where a low static strength resulted in lower stresses experienced during fatigue tests. These lower stresses limit crack growth and, therefore, help to create a good level of fatigue resistance [41]. The presence of defects such as pores or the misalignment of reinforcement in certain regions, as can be observed in some of the specimens in Figure 5b, could precipitate catastrophic failure in specimens. To a lesser extent, the presence of zones with a lower local reinforcement volume fraction could similarly affect the specimens.

To gain a better understanding of the damage suffered by the specimens after fatigue, the images taken by digital microscopy are exhibited below. Figure 7a shows a PA11 composite specimen after fatigue testing, revealing how the fracture progresses following the direction of the reinforcement. Regarding wettability, the contact angle between both polyamides and the reinforcement, around 145°, is very high, indicating a low wettability. This has an important effect during hot pressing: the capillary effect does not allow the polyamide to infiltrate between the rods when they are in close proximity, leaving weakened areas that are more prone to failure. Figure 7b illustrates how the crack propagates through the polyamide matrix into the contact between two rods, causing the matrix’s cracking but not fiber breakage.

Figure 8 shows the case of a PA12 composite specimen after the fatigue testing. White dashed circumferences indicate areas where the reinforcement has been exposed, indicating reinforcement–matrix debonding and reinforcement pull-out. Fiber breakage is not to be expected in these materials due to the relatively low strength of the thermoplastic matrix. Surface treatments such as atmospheric pressure plasma treatment could be applied to the reinforcements to improve reinforcement–matrix adhesion. Such improvements delay reinforcement–matrix debonding and result in stronger composite materials.

While the reinforcement distribution seemed homogeneous from what was observed in the μCT, the fiber orientation distribution seems critical when analyzing the tensile and fatigue results. Manufacturers of composites at a large scale should consider the importance of a one-dimensional nature when aiming to replicate these composites.

### 3.4. Thermomechanical Behavior

The real and imaginary parts of the complex modulus, E*, are the storage and loss modulus, E′ and E″, respectively. The loss modulus, denoted by the product of the storage modulus and the loss factor (tan δ), is a figure of merit typically utilized to characterize the damping qualities of materials.

#### 3.4.1. Thermomechanical Behavior of Polymers

Figure 9 presents the thermomechanical behavior of the polyamides employed as matrices. Figure 9a,b represent the storage modulus as a function of temperature for PA11 and PA12, respectively. In the same manner, Figure 9c,d depict the loss modulus as a function of temperature for PA11 and PA12, respectively. The storage and loss modulus peaks are good indicators of the structural changes in the polymers. The significant peak found in the loss modulus indicates the presence of T_g_, occurring at an α-relaxation temperature. A smaller peak, located at temperatures near −15 °C in the loss modulus curves, is indicative of structural relaxations during molecular rotations, corresponding to the β-relaxation [42].

The temperature indicated by the inflection point in the storage modulus curve can be used to determine the T_g_. The inflection point is determined as the intersection of the glassy and transition regions. However, for the sake of representativeness, the maximum of the loss modulus has been taken as indicative of T_g_. Table 6 displays the storage modulus value at the inflection point for storage modulus, the peak value of the loss modulus, and the T_g_ determined in both of these manners.

The T_g_ obtained by both the storage and loss moduli is consistent with those reported in the literature [29], with the T_g_ of PA11 being slightly higher. Comparing PA11 against PA12, the storage modulus around the T_g_ is shown to be 11% higher, while the value of the storage modulus at room temperature (23 °C) is 11% higher. Considering the loss moduli around the T_g_, the values are rather similar, with the one for PA11 being 3% higher.

#### 3.4.2. Thermomechanical Behavior of Composites

Figure 10 presents the thermomechanical behaviors of the polyamide composites. Figure 10a,b exhibit the storage modulus as a function of temperature for the PA11 and PA12 composites, respectively. Figure 10c,d exhibit the loss modulus as a function of temperature for the PA11 and PA12 composites, respectively.

Considering the evolution of the storage modulus for the polyamide composites, there is not such an abrupt change from the glassy to the rubbery state as in the previous case of the polymers due to the presence of the reinforcement. Two inflection points and two peaks are observed in the storage modulus and loss modulus plots, respectively, in the case of the composites in Figure 10c,d. The lower temperature point corresponds to the α-transition, T_g_ of the polyamide matrix, while the upper temperature reported corresponds to the T_g_ of the epoxy, the reinforcement matrix, reported in Table 2. Table 7 presents the values corresponding to the two aforementioned points, considering the storage modulus and loss modulus, together with their CoV.

The inherent heterogeneity of the recycled composites is evident from the CoV in Table 7, especially for the PA11 composites. A difference in the storage modulus of almost 4 GPa can be explained by V_f_ effects. In other words, the divergence is related to the number and position of the composite rods in the fracture area, rather than being due to the change of the matrix. For the PA11 composites at room temperature, 23 °C, the storage modulus is 32% higher. At the temperature corresponding to the polyamide T_g_, the storage modulus is 34% higher, and at the temperature of the T_g_ of the epoxy, the second inflection point, the storage modulus is 43% higher for the PA11 composites. Considering the loss modulus, the PA11 composites exhibit higher values compared to the PA12 composites. Particularly, the loss modulus value near the first peak, corresponding to the polyamide T_g_, is 21% higher, while the value of the second peak, compared to the epoxy T_g_, is 53% higher in the case of the PA11 composites.

Considering the effect of the presence of the reinforcement in the T_g_ of the polyamides and manipulating the data from Table 6 and Table 7, it is observed that it increases in both cases. For the PA11 composites, the T_g_ has increased by 5.3 °C according to the storage modulus and by 5.8 °C according to the loss modulus. For the PA12 composites, the T_g_ is 3.1 °C and 4.8 °C higher than in the case of the polymers, according to the storage and the loss modulus, respectively. This effect can be due to the presence of carbon fibers and epoxy, most likely affecting the crystallinity. A fast cooling rate would have caused a lower crystallinity, thereby delaying the T_g_.

## 4. Conclusions

In this research paper, we developed and carried out a mechanical, dynamic, and thermomechanical characterization of two composites, consisting of a thermoplastic matrix reinforced with mechanically recycled long carbon fibers. The thermomechanical behavior of the composites was determined—with a similar one for the polymers—to not only compare composites against each other but against the polymers themselves. The following conclusions can be drawn:The process developed to manufacture a new thermoplastic composite incorporating carbon fiber-reinforced epoxy composite waste rods as fillers has been successfully carried out, enabling the study of its properties.The V_r_ and V_f_ were calculated from theoretical values and from the analysis of μCT images. The results, complemented with a statistical analysis, were satisfactory, showing that the real values are close to the theoretical values.The tensile strength and tension–tension fatigue tests exhibited the inherent heterogeneity of the recycled composites, reflected in the large coefficients of variation. The tensile strength test results barely indicated any difference between the composites. Nonetheless, the PA11 composites presented a less stiff behavior, exhibiting a higher strain to failure.Digital microscopy after the fatigue testing showed that the fracture follows the fiber orientation to find the easiest path. In the fatigue process, the crack propagated through the polyamide matrix and caused the matrix’s cracking. Regarding the damage analysis, matrix cracking, reinforcement–matrix debonding, and reinforcement pull-out were observed.PA12 exhibited a better behavior than PA11, as the storage modulus was 11% higher around T_g_. The T_g_ of PA11 (49.9 °C) was found to be higher than that of PA12 (44.6 °C).Considering the composites, the PA11 composites exhibit higher storage moduli than PA12, which were 34% higher around polyamide T_g_ and 43% higher around epoxy T_g_. The T_g_ is altered for both composites, increasing by 5.3 °C and by 5.8 °C according to the storage and loss moduli, respectively, for the PA11 composites. For the PA12 composites, the T_g_ is 3.1 °C and 4.8 °C higher than in the case of the polymers, according to the storage modulus and the loss modulus, respectively.

Our future research is aimed toward the recycling and reuse of end-of-life or expired aeronautical and automotive composites without removing their previous matrices and their incorporation in a bio-based polyamide to achieve sustainable materials.

## Figures and Tables

**Figure 1 polymers-14-03951-f001:**
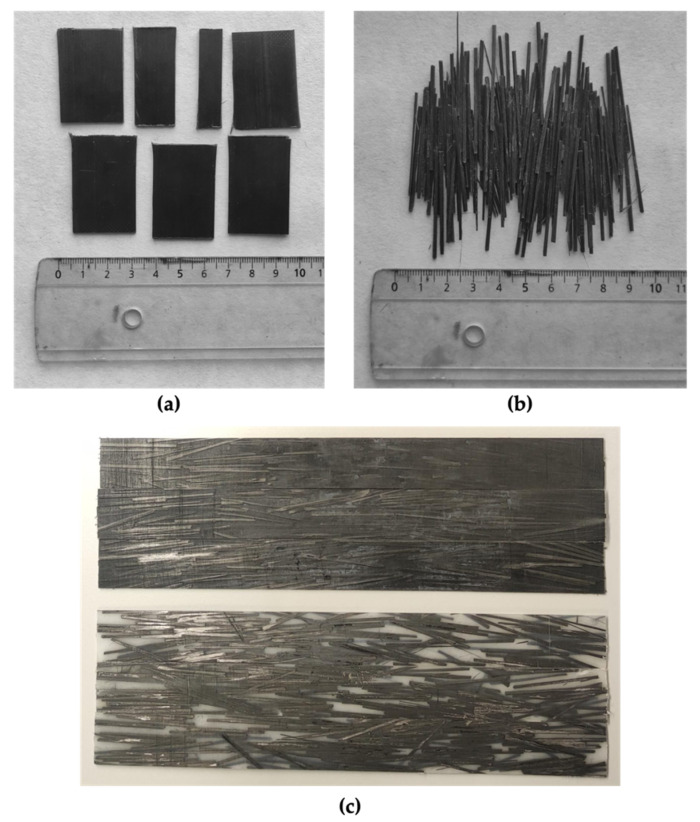
(**a**) Carbodur S512 before cutting. (**b**) Carbodur S512 after cutting. (**c**) Manufactured composites: PA11 (**top**) and PA12 (**bottom**).

**Figure 2 polymers-14-03951-f002:**
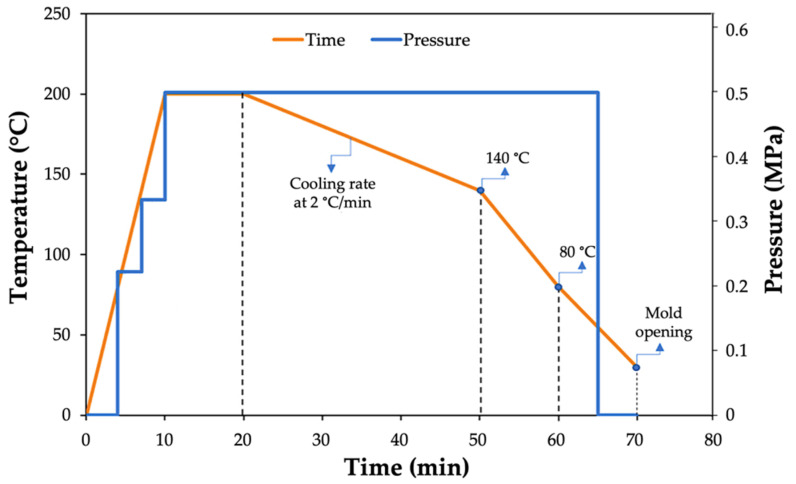
Hot press cycle for manufacturing PA matrices and PA composites. Adapted from Ref. [29].

**Figure 3 polymers-14-03951-f003:**
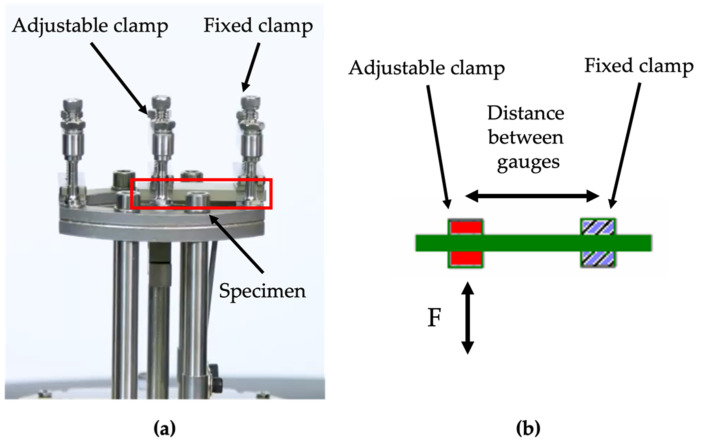
(**a**) Image of DMA single cantilever clamp. (**b**) Schematic of DMA single cantilever clamp.

**Figure 4 polymers-14-03951-f004:**
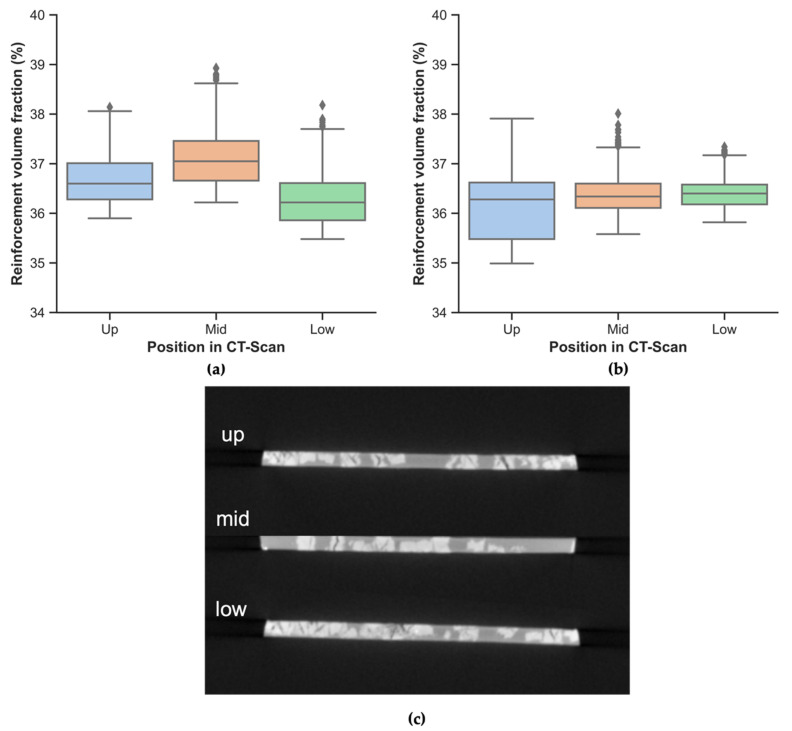
Reinforcement volume fraction (V_r_) obtained by μCT. (**a**) PA11 composites; (**b**) PA12 composites. (**c**) μCT image from PA11 composites’ cross-section before applying thresholding. The line in the middle of the box in (**a**,**b**) corresponds to the median; the upper and lower limits of the box correspond to the 75th and the 25th percentile, respectively. Whiskers extend beyond the limits of the box to a maximum of 1.5 times the interquartile range (IQR), the difference between the 75th and the 25th percentile. Outliers are shown as individual points beyond the whiskers. In (**c**), the background is black, the matrix is grey, and the reinforcement is whitish.

**Figure 5 polymers-14-03951-f005:**
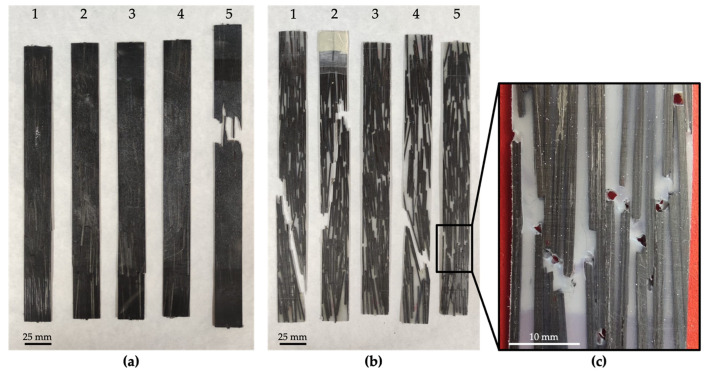
Tensile test specimens. (**a**) PA11 composites, (**b**) PA12 composites, and (**c**) detail of the failure area of a PA12 composite specimen. The color difference is due to the fact that PA11 is black, while PA12 is white.

**Figure 6 polymers-14-03951-f006:**
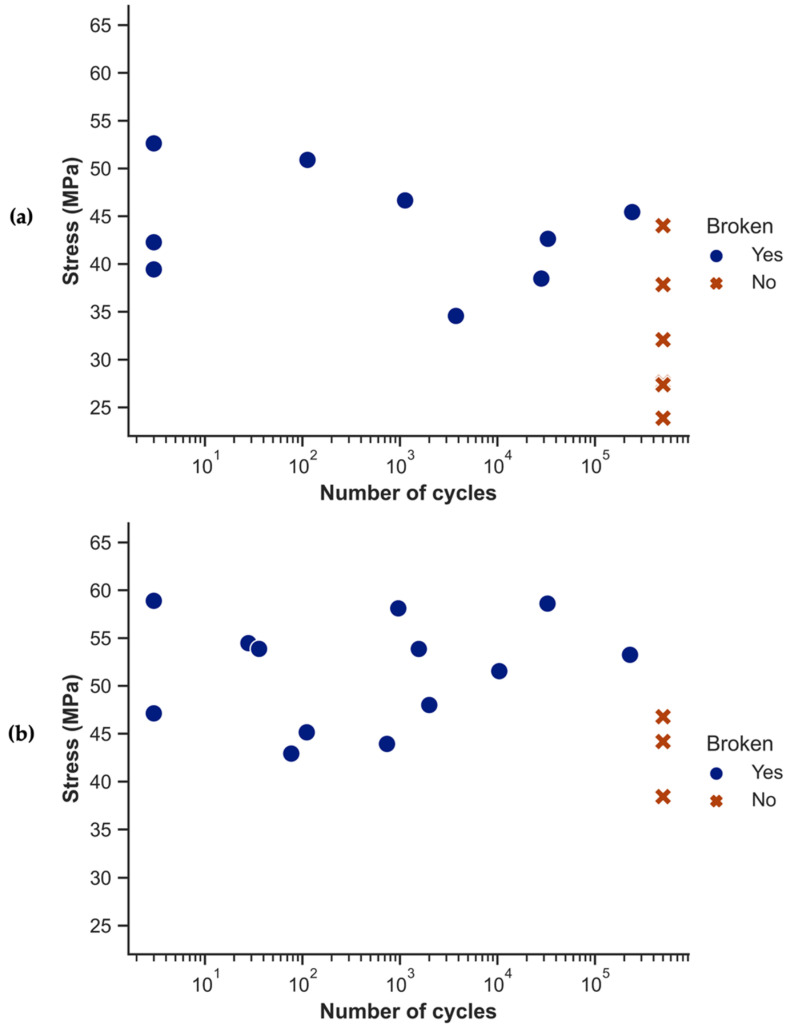
S-N curves obtained after tension–tension fatigue testing. (**a**) PA11 composites; (**b**) PA12 composites.

**Figure 7 polymers-14-03951-f007:**
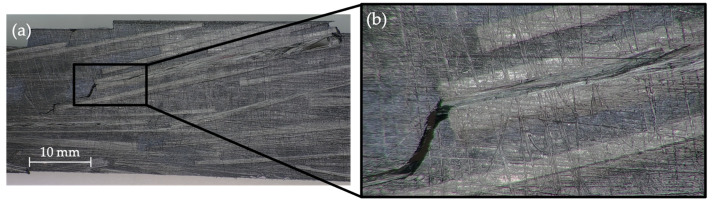
Digital microscopy images of a PA11 composite after fatigue. (**a**) magnification of 10×; (**b**) magnification of 50×.

**Figure 8 polymers-14-03951-f008:**
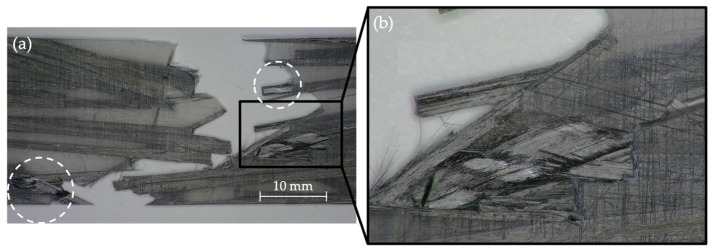
Digital microscopy images of a PA12 composite after fatigue. (**a**) magnification of 10×; (**b**) magnification of 50×. White dashed circumferences point to areas where the reinforcement has been exposed.

**Figure 9 polymers-14-03951-f009:**
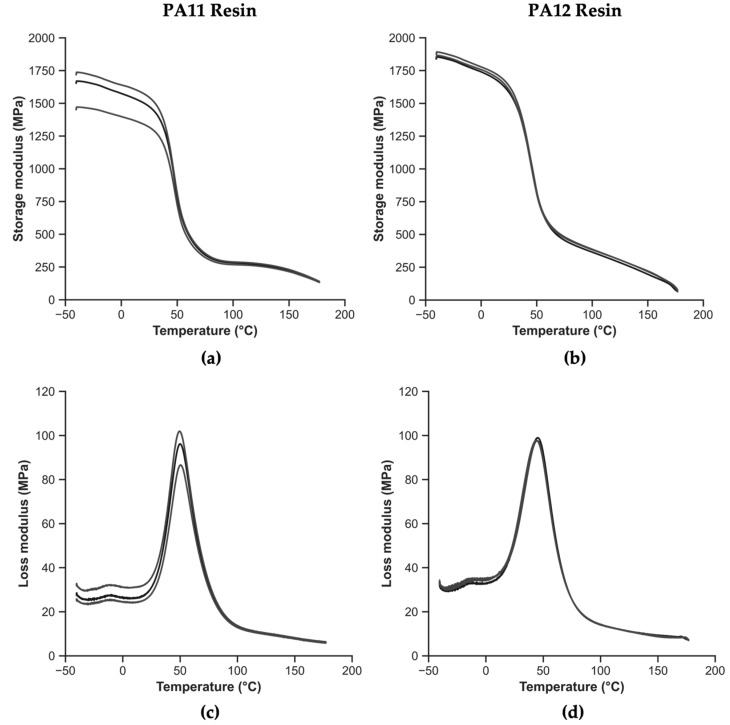
Thermomechanical behavior of polyamides: (**a**) storage modulus as a function of temperature for PA11, (**b**) storage modulus as a function of temperature for PA12, (**c**) loss modulus as a function of temperature for PA11, and (**d**) loss modulus as a function of temperature for PA12.

**Figure 10 polymers-14-03951-f010:**
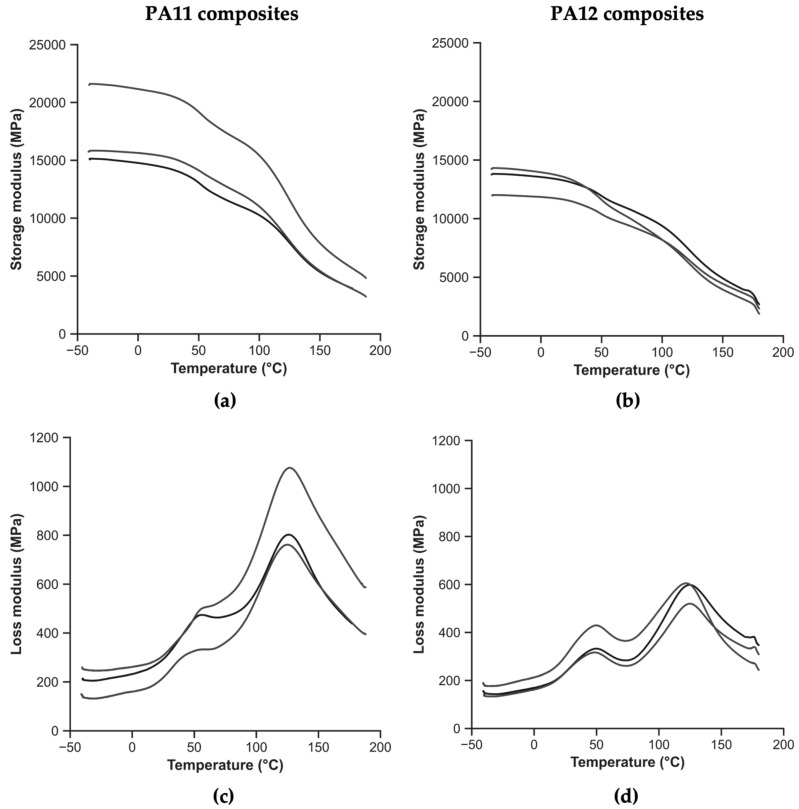
Thermomechanical behavior of composites: (**a**) storage modulus as a function of temperature for PA11 composites, (**b**) storage modulus as a function of temperature for PA12 composites, (**c**) loss modulus as a function of temperature for PA11 composites, and (**d**) loss modulus as a function of temperature for PA12 composites.

**Table 1 polymers-14-03951-t001:** Properties of the matrices and the reinforcement.

Material	Density (g/cm^3^)	Glass Transition Temperature (°C)	Melting Point (°C)	Longitudinal Modulus (GPa)	Transverse Modulus (GPa)	Longitudinal Poisson’s Ratio	In-Plane Shear Modulus (GPa)	Ref.
PA11	1.05	45.95	188.50	1.76	1.76	0.35	0.63	[29,30]
PA12	1.01	49.29	178.05	2.96	2.96	0.35	1.07	[29,31]
Carbodur S512	1.60	>100	-	165	9	0.28	5	[32,33]

**Table 2 polymers-14-03951-t002:** Technical information of Carbodur S512 as reported in the Product Data Sheet [33], according to EN 2561.

Material	Fiber Volume Fraction (%)	Laminate Thickness (mm)	Tensile Strength (MPa)	Elastic Modulus (GPa)	Strain at Failure (%)	Ref.
Carbodur S512	>68	1.2	2900	165	1.80	[32]

**Table 3 polymers-14-03951-t003:** Tests carried out, specimen dimensions, and standards used for testing.

Test	Specimen Type	Polymer or Matrix Material	Distance between Gauges (mm)	Width (mm)	Thickness (mm)	Standard
Tensile strength	Composite	PA11	150.0 ± 0.1	24.9 ± 0.2	1.23 ± 0.03	ISO 527-5
		PA12		25.1 ± 0.1	1.40 ± 0.02	
Fatigue	Composite	PA11	150.0 ± 0.1	24.9 ± 0.1	1.27 ± 0.06	ISO 527-5
		PA12		25.0 ± 0.1	1.43 ± 0.04	
DMA	Polymer	PA11	17.8 ± 0.1	15.6 ± 0.4	2.02 ± 0.10	ISO 6721-11
		PA12		15.4 ± 0.4	2.11 ± 0.02	
	Composite	PA11		16.4 ± 0.4	1.19 ± 0.02	
		PA12		14.2 ± 0.3	1.43 ± 0.06	

**Table 4 polymers-14-03951-t004:** V_r_ and V_f_ obtained by means of the rule of mixtures in composites and by the analysis of X-ray micro-computed tomography.

Material	Method	Reinforcement Volume Fraction (V_r_) (%)	Fiber Volume Fraction (V_f_) (%)
PA11 composites	Rule of mixtures	36.5	24.8
	μCT	36.7 ± 0.6	24.9 ± 0.4
PA12 composites	Rule of mixtures	37.4	25.4
	μCT	36.3 ± 0.5	24.7 ± 0.3

**Table 5 polymers-14-03951-t005:** Tensile properties of PA11, PA12, and the composites manufactured with both polyamides. Tensile properties of PA11 and PA12 are reported in [29].

Material		Ultimate Tensile Strength (MPa) [CoV]	Young’s Modulus (GPa) [CoV]	Strain at Failure (%) [CoV]
Polymers	PA11	20.0 ± 3.8 [19.0]	1.7 ± 0.1 [7.8]	4
	PA12	64.0 ± 3.2 [5.0]	3.0 ± 0.1 [3.5]	200
Composites	PA11	106.1 ± 10.1 [9.5]	21.4 ± 1.2 [10.0]	0.9 ± 0.1 [5.4]
	PA12	109.9 ± 12.2 [11.1]	23.2 ± 2.9 [8.1]	0.7 ± 0.1 [12.4]

**Table 6 polymers-14-03951-t006:** Inflection points for storage modulus and peak value for loss modulus for the polymers and their associated temperatures.

Polymer	Storage Modulus		Loss Modulus	
	Value at Inflection Point (MPa) [CoV]	Point Temperature (°C) [CoV]	Peak Value (MPa) [CoV]	Peak Temperature (°C) [CoV]
PA11	900 ± 70 [8.2]	47.5 ± 0.4 [0.9]	95 ± 8 [8.1]	49.9 ± 0.4 [0.8]
PA12	1010 ± 30 [3.0]	45.8 ± 0.9 [1.9]	98 ± 1 [0.7]	44.6 ± 0.5 [1.2]

**Table 7 polymers-14-03951-t007:** Inflection points for storage modulus and peak values for loss modulus for composites and their associated temperatures, considering two interesting inflection points and peaks.

Composite	Storage Modulus				Loss Modulus			
	First Inflection Point		Second Inflection Point		First Peak		Second Peak	
	Value (MPa) [CoV]	Temperature (°C) [CoV]	Value (MPa) [CoV]	Temperature (°C) [CoV]	Value (MPa) [CoV]	Peak Temperature (°C) [CoV]	Value (MPa) [CoV]	Peak Temperature (°C) [CoV]
PA11	15,200 ± 3200 [21.1]	52.8 ± 1.0 [1.8]	9500 ± 2000 [21.4]	122.8 ± 2.5 [2.0]	430 ± 90 [20.3]	55.7 ± 1.1 [1.9]	880 ± 170 [19.4]	125.7 ± 0.4 [0.3]
PA12	11,400 ± 800 [7.3]	48.9 ± 0.4 [0.8]	6600 ± 600 [8.9]	122.7 ± 0.2 [0.1]	360 ± 60 [17.0]	49.4 ± 0.9 [1.8]	570 ± 50 [8.2]	123.8 ± 1.8 [1.4]

## Data Availability

Not applicable.

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
