# Peer review of "Novel Thermoplastic Composites Strengthened with Carbon Fiber-Reinforced Epoxy Composite Waste Rods: Development and Characterization"

_polymers, 2022, doi:10.3390/polym14193951_

Round 1
Reviewer 1 Report
The present manuscript made a study on the manufacturing of two novel composites, to evaluate the fiber distribution and to perform a mechanical, dynamical, and thermomechanical characterization. The logicality and accuracy of this article are satisfied. And several experiments were conducted to obtain the main data of those composites. However, some problems should be mentioned, some pieces of advice are listed as follows:
1. why did the author select PA11 and PA12 as matrices of novel composites?
2. In column 106 some words are missing.
3. For all experimental tests, the author should add some figures about the specimen and equipment, which make the results more reliable.
4. The y-axis of figure 7 (a) and (b) should be storage modulus?
5. Figure 5 and 6 should show more details. 10x and 50x show no differences and cannot get more conclusions.
Author Response
Dear reviewer, please see the attachment.
Faithfully yours,
J. A. Butenegro.

Reviewer 2 Report
The authors conducted the comprehensive experimental research to reuse carbon-fiber reinforced thermoplastic composite wastes as reinforcing phase of a new secondary thermoplastic composites. The research is within the scope of the Journal with clear research objectives, reasonings, new experimental design and related sample fabrication, tests, and data reduction and discussions. The research well presents a practical direction for energy-efficient recycling high-grade composites for use as reinforcing phase of secondary new composites. The paper can be accepted for publication in this Journal while a few minor revisions can be further made to further enhance the quality of the paper as follows:
1. The title could be: Novel thermoplastic composites strengthened with whiskers of carbon-fiber reinforced polymer composite wastes: Development and characterization.
2. This reviewer suggests the authors to use "whiskers" but not "sticks" throughout the paper as the former term has been broadly accepted in composites community while the latter is unfamiliar and not professional term somehow.
3. In Line 105, "a new carbon fiber stick-reinforced thermoplastic composite" is off the original meaning of the paper: carbon-fiber polymer composite whiskers, but not "carbon fiber sticks..."
4. In Section of Material (Line 115), PA11 and PA12 in context can be typically termed as "polymeric resins" in composite processing but not just as "thermoplastics."
5. According to the paper, the whiskers (sticks herein) carried the width of 1.0 to 1.5 mm, while the formed PA11 and PA12 composite specimens carried the width of ~1.2 to 2 mm, which means the whiskers were mainly distributed in the planar way while not in the random three-dimensional distribution. The authors need to address this very special planar case as these composites could not be considered as the general three-dimensional composites, nor unidirectional composites. In addition, the authors are suggested to estimate the optimal or practical whisker length according to a simple shear-lag model, by which the whisker length can be optimized and could be further shortened theoretically (The authors may place this part in discussion while not further adding new tests. In the next paper, the authors could take into account this effect). In this way, the stress state can be improved and the processing condition can also be improved such as extrusion. The resulting mechanical properties can be further enhanced since long-fiber composites exist strong stress anisotropicity somehow.
7. Though the authors ignored, it is technically very valuable to characterize and discuss the surface morphology of the whiskers (sticks herein) on the strengthening effect of the resulting composites. The authors are suggested to add a few sentences in Discussions to show their knowledge on it though they have not investigated it in this paper.
8. "six random specimens" in Line 181 could be changed as "six randomly selected specimens."
9. "Young's Modulus" should be "Young's modulus" throughout the paper. The authors are suggested to use "Search" and "Replace" to change.
10. In Line 319, the beginning of the caption of Figure 3 can be "Tensile test specimens,"
11. In Line 328, this sentence could be revised as "Catastrophic failure happens when these pores and cavities are penetrated by the simpleset failure paths."
12. The S-N curves do not have the typical S-N curves of composites in Figure 4, which means a few fabrication-related failure mechanisms have not exhausted, which also show the potential improvement for such composites. From the paper, the processing of the composites is pretty rough and the composites could be controlled in a good manner in terms of whisker orientation, whisker/matrix distribution, and whisker surface morphology (partial in carbon fiber and partial in epoxy), and so on. It is good for the authors to point out the main factors to influence the S-N curves or the earlier failure specimens.
13. In Figure 5, the 145-degree failure in matrix was due mainly to the large deformation of the compliant resin that was constrained between two misaligned, stiff whiskers with nearly rigid motion under uniaxial tension.
14. In Figures 7 and 8, the authors need to add the figure legends to indicate the three specimens used in each figure, otherwise they look misleading.
15. "PA11" and "PA12" in Figure 7 could be changed into "PA11 Resin" and "PA12 Resin", while "PA1" and "PA12" in Figure 8 could be changed into "PA11 Composite" and "PA12 Composites" to clearly show their differences from the former.
16. In Conclusions, the authors are suggested to remove "a more powerful method, such as scanning electron microscopy (SEM) ...." as nowadays, use of SEM in composite research is a very routine technical process. Not using SEM in this paper is the authors' default and technical shortcomings in this research paper, just not mentioning it or mentioning ir in Discussion but not in Conclusions.
Author Response

(The authors gave the same response as above.)

Reviewer 3 Report
It's good research paper, only the minor revisions are required, namely:
1) Section 2.2. It would be better to illustrate the section with photos (outlook) of raw materials (including cutted sticks of Carbodur) and final experimental samples.
2) Section 3.1.2. Please illustrate the section with several mictoCT images.
3) Figure 4. Please give the difference between the broken and not broken samples not only by color (for instance, broken samples can be indicated as open circles or as square)
Author Response

(The authors gave the same response as above.)

Round 2
Reviewer 1 Report
The paper after the clarifications is suitable for publication.